

# Comparison of mid-latitude single and mixed-phase cloud optical depth from co-located infrared spectrometer and backscatter LiDAR measurements

Gianluca Di Natale[1], Marco Barucci[1], Claudio Belotti[1], Giovanni Bianchini [1], Francesco D'Amato[2], Samuele Del Bianco[2], Marco Gai[2], Alessio Montori[1], Ralf Sussmann[3], Silvia Viciani[1], Hannes Vogelmann[3], and Luca Palchetti[1]

[1]National Institute of Optics, Via Madonna del Piano 10, Sesto Fiorentino, Firenze, Italy
[2]Institute of Applied Physics, Via Madonna del Piano 10, Sesto Fiorentino, Firenze, Italy
[3]Karlsruhe Institute of Technology, IMK-IFU, Garmisch-Partenkirchen, Germany

**Correspondence:** Gianluca Di Natale (gianluca.dinatale@ino.cnr.it)

**Abstract.** The longwave downwelling spectral radiance measurements performed by means of the Far-Infrared Radiation Mobile Observation System (FIRMOS) spectrometer at the summit of the Mt. Zugspitze (German Alps) in the Winter 2018–2019, allowed to retrieve the optical and micro-physical properties of ice, mixed and water clouds, showing a good agreement of the statistical relationship between the ice water path and the ice optical depth with the ones from previous works. In this

paper the optical depths retrieved from FIRMOS are initially compared with selected cases calculated from backscattering Light Detection And Ranging (LiDAR) data by using a transmittance method. Then, in order to compare the whole FIRMOS dataset, the power-law relationship between backscattering and extinction is used to apply the Klett method and automatize the routine. Minimizing the root mean square differences, the exponent of the relationship, the so called backscatter-extinction coefficient ratio, is assessed to be 0.85 with a variabiliy in the range 0.60–1.10 for ice clouds and 0.50 with a variability within

0.30–0.70 for mixed and water clouds.

## 1   Introduction

The assessment of cloud radiative properties, particularly of cirrus and mixed clouds, is a paramount objective to improve the accuracy of general circulation models (GCM) (Lubin et al., 1998) and it represents an outstanding problem for the remote sensing and radiative transfer communities. To date, there is still a lack of measurements of longwave spectral properties of

cirrus clouds, despite the fact that they represent one of the key component of the Earth Radiation Budget (ERB) (Cox et al., 2010; Kiehl and Trenberth, 1997). As a consequence, the uncertainties in the parameterization of their radiation properties are still large (Yang et al., 2015).

Clouds with temperature between 0 °C and -38 °C, where particles can either be frozen or liquid, represent the so called mixed-phase clouds and their degree of glaciation depicts one of the largest uncertainties in climate prediction as stated by

Costa et al. (2017), there are in fact many gaps that still remain in the experimental and theoretical description of this type of clouds (Korolev et al., 2017). Within this temperature regime, clouds can occur as liquid only (supercooled water) as well





as in mixed-phase, in which ice crystals coexist with water droplets, and also as purely ice. Mixed-phase clouds are very common in polar regions, both in Artic (Turner, 2003) and Antarctica (Cossich et al., 2021), but they can either be widely found at mid-latitude, so much that Costa et al. (2017) shows that out of 16 measurements flights of the COALESC (Combined
Observation of the Atmospheric boundary Layer to study the Evolution of StratoCumulus) campaign performed with aircraft at mid-latitude, 14 observations provided the presence of clouds in the mixed-phase regime.

Ice and mixed-phase clouds exert a very strong effect on the spectral radiance in the Mid and Far Infrared (MIR, FIR) portion, between 100-1600 $cm^{-1}$ (6-100 $\mu m$ ), since they modulate the incoming solar radiation in the shortwaves and the outgoing thermal emission at longwaves coming from the ground and lower atmosphere. For this reason, our knowledge about the cloud
radiative properties has to be improved starting by increasing satellite, ground and in situ observations.

In September 2019 the Far infrared Outgoing Radiation for Monitoring and Understanding (FORUM) was selected by the European Space Agency (ESA) as the ninth Earth Explorer (EE-9), with the purpose of studying the FIR portion of the top-of-atmosphere (TOA) Earth's emission spectrum, below 667 $cm^{-1}$ (wavelengths above 15 $\mu m$ ) (Palchetti et al., 2020b). The FORUM measurement will cover the broad band 100-1600 $cm^{-1}$ with a spectral resolution of 0.5 $cm^{-1}$ (full width half
maximum), opening the capability to improve the knowledge of the FIR spectral properties of water vapour and cirrus clouds, whose radiative effect depends on their optical and micro-physical properties, such as the particle effective sizes and crystal habits.

During the so called Phase-A (feasibility study) of the satellite development, the Far-Infrared Radiation Mobile Observation System (FIRMOS) project was started with the aim of deploying a prototype of the FORUM spectrometer to perform on field
spectral measurements of the atmospheric downwelling longwave radiation (DLR) with ground-based observations. Spectral measurements of DLR, up to the FIR region, are already successfully exploited to retrieve cloud properties, in particular cirrus clouds, at mid-latitude (Palchetti et al., 2016; Maestri et al., 2014) and polar regions (Di Natale et al., 2017; Maesh et al., 2001; Turner, 2003; Garrett and Zhao, 2013; Rowe et al., 2019). DLR measurements are, in general, very important since while they provide the complementary part of the TOA spectral radiance, they also form a very useful basis to test the retrieval algorithms
for the future satellite applications, such as FORUM.

In this paper, the cloud optical and micro-physical properties are simultaneously retrieved, together with the thermodynamic phase (ice, water or mixed) using FIRMOS measurements. The resulting optical depths are in accordance with those obtained from selected measurements of a co-located backscatter LiDAR. Finally, the relationship among the retrieved optical depth and the ice water path is shown and compared with the results presented in previous works.
FIRMOS and LiDAR observations and the data analysis methodology are described in section 2, the results are presented and discussed in section 3 and the conclusions are drawn in section 4.

## 2 Observations and methodology

The FIRMOS spectrometer was deployed from December 2018 to February 2019 to perform systematic measurements of the longwave downwelling spectral radiance emitted by the atmosphere Palchetti et al. (2020b) at the Alpine observatory on the





summit of Mt. Zugspitze (47.421 °N, 10.986 °E) in the South of Germany, at 2962 m a.s.l. The instrument acquires spectra in the broad band between 100–1000 cm$^{-1}$ with an unapodized spectral resolution of 0.3 cm$^{-1}$ every 8 minutes. A backscatter stratospheric aerosol LiDAR operating in semi-automatic mode at a wavelength of 532 nm was located at the Environmental Research Station Schneefernerhaus on the south slope of the Mount, 300 m below (2675 m a.s.l.) and 680 m southwest from the summit; it operated with a frequency of one profile every 4-10 minutes with an integration time of 1 min and a vertical

resolution of 7.5 m.

## 2.1 Spectral measurements and retrieval of cloud parameters

The retrieval of atmospheric state and cloud parameters is performed by exploiting the band between 200–1000 cm$^{-1}$ (10–50 $\mu$m ), which provides the most relevant information about water vapour, temperature and cloud properties. The atmospheric window between 820–980 cm$^{-1}$ and the microwindows below 600 cm$^{-1}$ provide a high signal coming from the clouds; the

band falling in the region 200–600 cm$^{-1}$ provides information about the water vapour profile and the CO$_2$ band between 600–750 cm$^{-1}$ gives us information about the atmospheric temperature. The Simultaneous Atmospheric and Clouds Retrieval (SACR) algorithm (Di Natale et al., 2020), which includes a forward and a retrieval model based on the optimal estimation (OE) approach, is used to retrieve simultaneously the cloud optical and micro-physical properties, such as the effective diameter of the ice particles and water droplets, the optical depth and the ice fraction, together with the atmospheric profiles of water

vapour and temperature.

The effective diameter ($D_{ei}$) of the ice crystals is defined following (Yang et al., 2005):

$$D_{ei} = \frac{3}{2} \frac{\sum_{h=1}^{N} \int_{L_{min}}^{L_{max}} f_h(L) V_h(L) n(L) dL}{\sum_{h=1}^{N} \int_{L_{min}}^{L_{max}} f_h(L) A_h(L) n(L) dL} \tag{1}$$

where $L$ is the maximum length of the ice crystals, $n(L)$ is the particle size distribution, $V_h$ and $A_h$ represent the particle volume and the projected area, respectively, the pedix $h$ denotes the habit index and $f_h$ indicates the habit fraction defined such

that $\sum_{h=1}^{N} f_h = 1$ at each particle length $L$ and $N$ is the total number of habits. For water droplets, assuming the spherical shape, the optical properties are derived from the Mie theory and eq. (1) reduces to the effective diameter $D_{ew}$:

$$D_{ew} = 2 \frac{\int_{r_{min}}^{r_{max}} r^3 n(r) dr}{\int_{r_{min}}^{r_{max}} r^2 n(r) dr} \tag{2}$$

where $r$ is the radius of the droplets. The particle size distribution is modeled with a gamma size distribution both for ice and water as done by Turner (2005).

The issue of the treatment of mixed-phase clouds is simplyfied by assuming that mixed-phase clouds composed of a uniform mixture of spherical water droplets and ice crystals, which in turn are modeled as a mixture of different habits. This assumption is made since no depolarization measurements to discriminate the vertical distribution of thermodynamic phase, were available. However, in order to take into account the multiple scattering effect at different heights and the layers inhomogeneity, the





normalized LiDAR raw signal is used to modulate the internal distribution of OD within the cloud (Di Natale et al., 2020). The

different behaviour of the ice and water refractive index below 1000 cm$^{-1}$ (Turner, 2005) is exploited to discriminate the solid and liquid component from the infrared spectrum.

The total ice fraction ($\gamma$) is defined as the ratio between the Ice Water Path (IWP) and the Total Water Path (TWP) (Yang et al., 2003) as follows:

$$\gamma = \frac{\text{IWP}}{\text{TWP}} \tag{3}$$

where the TWP = IWP + LWP is the sum of the ice and liquid component, respectively. $\gamma$ is fitted together with D$_{\text{ei}}$, D$_{\text{ew}}$ and the optical depth at visible wavelength (OD = OD$^{ice}$+OD$^{wat}$). This latter parameter is related to the optical depth in the FIR band at the wavenumber $\nu$ (OD$_{\text{FIR},\nu}$) by the equation:

$$\text{OD}_{\text{FIR},\nu} = \text{OD}\frac{\langle Q_{\text{e}} \rangle_{\text{FIR},\nu}}{\langle Q_{\text{e}} \rangle} \tag{4}$$

where $Q_{\text{e}}$ is the extinction efficiency. These parameters, together with the absorption efficiency, the single scatttering albedo

and the asymmetry factor for ice and water are mixed up in the radiative transfer solution as described in Yang et al. (2003).

The Instrument Line Shape (ILS) of the FIRMOS spectroradiometer is also fitted, in the same way as done with the REFIR-PAD spectroradiometer operating in Antarctica at Dome-C (Palchetti et al., 2015; Bianchini et al., 2019), as a linear combination of a sinc function, which generally contributes with more than 90 %, and a sinc$^2$ function, which allows to take into account the self-apodization due to the finite internal solid angle $\Omega$ of the instrument. The resulting ILS is expressed as follows:

$$\text{ILS}_\nu(\Omega) = \alpha_\nu(\Omega) \cdot \text{sinc}(\frac{\nu}{\Delta\nu}) + (1 - \alpha_\nu(\Omega)) \cdot \text{sinc}^2(\frac{\nu}{2\Delta\nu}) \tag{5}$$

where $\Delta\nu$ is the spectral resolution (equal to 0.3 cm$^{-1}$ ) and $\nu$ represents the wavenumber. The $\alpha$ coefficient depends on $\Omega$ as shown in Bianchini et al. (2019). $\Omega$ is also fitted together with a frequency scale shift $\beta$, which takes into account both the shift due to the finite aperture given by $\nu(1 - \frac{\Omega}{4\pi})$ (Davis et al., 2001), and the effect due to the instability of the reference laser, as follows (Palchetti et al., 2016):

$$\nu' = (1 + \beta) \cdot \nu \tag{6}$$

On average values equal to 0.001 sr and $5 \cdot 10^{-5}$ are found for $\Omega$ and $\beta$. The state vector used in the retrieval is then given by:

$$\mathbf{x} = (D_{\text{ei}}, D_{\text{ew}}, \gamma, \text{OD}, \mathbf{U}, \mathbf{T}, \Omega, \beta) \tag{7}$$





where **U** and **T** represent the vectors of water vapour an temperature profiles fitted levels. The levels are taken at 2.962 km (corresponding to the ground), 3.3 km, 3.9 km, 4.4 km, 5 km and 7 km for water vapour and 2.962 km, 2.966 km, 3.2 km, 3.4 km, 3.7 km, 4.4 km and 7 km for temperature. This choice comes from sensitivty studies, particularly the very first level of temperature is introduced in order to take into account the strong variability just above the instrument.

The routine minimizes the cost function given by:

$$\chi^2 = (\mathbf{y} - \mathbf{F}(\mathbf{x}))^T \mathbf{S}_y^{-1} (\mathbf{y} - \mathbf{F}(\mathbf{x})) + (\mathbf{x} - \mathbf{x}_a)^T \mathbf{S}_a^{-1} (\mathbf{x} - \mathbf{x}_a) \tag{8}$$

where $\mathbf{y}$, $\mathbf{F}$, $\mathbf{x}_a$ indicate the vectors of the measurements, the forward model and the vector containing the a priori parameters, respectively. $\mathbf{S}_y$ denotes the Variance-Covariance Matrices (VCM) of the measurements and contains the noise, which is considered uncorrelated and given by the sum of square of the Noise Equivalent to Signal Ratio (NESR) and the calibration error. The $\mathbf{S}_a$ matrix represents the VCM of the a priori estimate $\mathbf{x}_a$ and it is composed of the VCM a priori for clouds, atmospheric profiles and instrumental coefficients. The OE approach minimizes the cost function in eq. (8) through the iterative formula given by (Rodgers, 2000):

$$\mathbf{x}_{i+1} = \mathbf{x}_i + [\mathbf{K}_i^T \mathbf{S}_y^{-1} \mathbf{K}_i + \lambda_i \mathbf{D}_i + \mathbf{S}_a^{-1}]^{-1} [\mathbf{K}_i^T \mathbf{S}_y^{-1} (\mathbf{y} - \mathbf{F}(\mathbf{x}_i)) - \mathbf{S}_a^{-1} (\mathbf{x}_i - \mathbf{x}_a)] \tag{9}$$

where $\lambda_i$ denotes the Levenberg-Marquardt damping factor at the iteration $i$, $\mathbf{K}_i$ represents the Jacobian matrix of $\mathbf{F}$ and $\mathbf{D}_i$ is a diagonal matrix which is described in detail in (Di Natale et al., 2020). The error of the retrieved parameters can be calculated when the convergence is reached, when variations on $\chi^2$ are less than 1 ‰, through the formula (Rodgers, 2000):

$$\mathbf{S}_x = (\mathbf{K}^T \mathbf{S}_y^{-1} \mathbf{K} + \mathbf{S}_a^{-1})^{-1} \tag{10}$$

The cloud top and bottom heights (CTH and CBH) are fixed in the radiative transfer calculations and are inferred from the LiDAR backscattering signal interpolated on the FIRMOS acquisition times. The optical properties of cirrus clouds are tabulated for different crystal habits in specific databases (Yang et al., 2013). In this work a mixture of different habits typical for mid-latitude is used in eq. (1), as discussed in previous works (King et al., 2004). According to this for ice crystals length lower than 70 $\mu$m the habit distribution is composed of 50 % bullet rosettes, 25 % plates and 25 % hollow columns, while for crystals length greater than 70 $\mu$m the habit distribution is given by 30 % aggregates, 30 % of bullet rosettes, 20 % of plates and 20 % of hollow columns.

As an example of the retrieval, Fig. 1 shows the comparison of a FIRMOS spectrum (black) detected during the passage of a cirrus cloud at 7.5 km of altitude on the day 6 February 2019 and the simulated spectrum (red). The fit procedure is initialized with the CBH and CTH derived from the LiDAR Range Corrected Signal (RCS) detected by the backscatter LiDAR (in black in the left panel of Fig. 2) and the a priori profiles (in red in the central and right panel of Fig. 2) of water vapour and temperature provided by the National Centers for Environmental Prediction (NCEP), with error bars assumed equal to 50 % and 0.1 %,





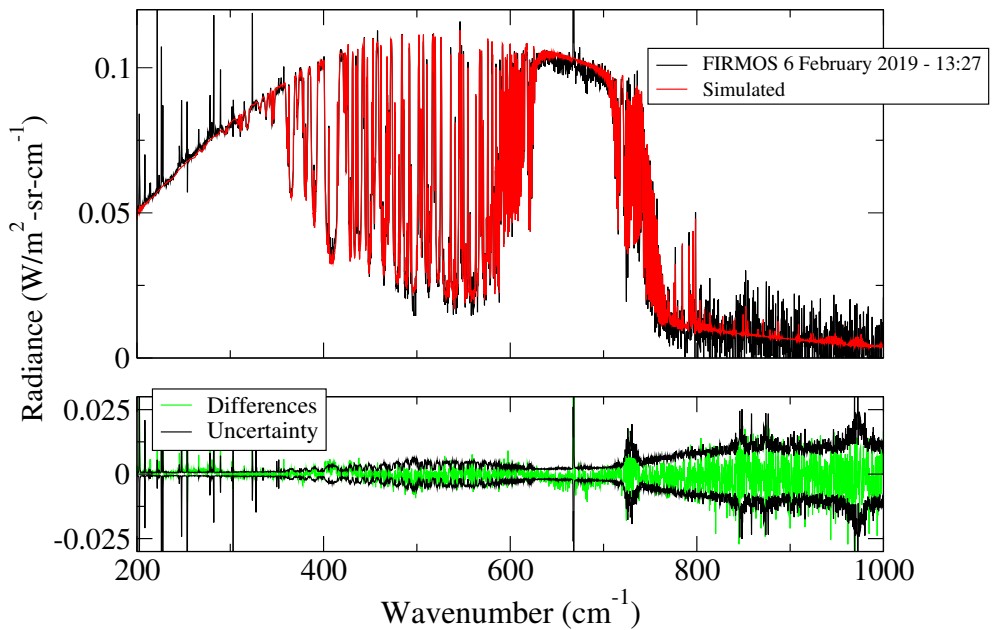

**Figure 1.** In the upper panel the comparison of the FIRMOS measurement (black) in the presence of a cirrus cloud at 7.5 km on 6 February 2019 at 13:27 with the simulated spectrum (red) at the last fit iteration. In the lower panel the differences (green) between the measurement and simulation in comparison with the instrumental uncertainty (black).

respectively, and assuming correlation lenghts equal to 2 km. Regarding the clouds, first guesses and a priori values are taken from previous studies with error equal to 100% in order to avoid an over-constraining of the retrieval procedure. Anyway, in

case the cloud bottom temperature goes down below -35 °C the $\gamma$ is automatically set to 0.95 with a stringent a priori error of 0.05, since at this temperature the coexistence of liquid water is very unlikely. The reduced $\chi^2$ turns out to be 1.18 and an effective diameter of (28.9 ± 4.5) $\mu$m and ice OD equal to (0.42 ± 0.04) are provided. The comparison of the radiance differences (green) are also reported in the lower panel of Fig. 1 together with the instrumental noise (black). In Fig. 2 the retrieved profiles (in blue) with the bars of the retrieval errors given by eq. (10), are also reported.

From the whole FIRMOS measurements a dataset of 245 spectra between 22 January and 7 February 2019 are selected within the time range of the acquired LiDAR measurements. From the analysis of these spectra 174 were ice clouds ($\gamma \geq 0.8$), 61 mixed clouds ($0.3 < \gamma < 0.8$) and 10 water clouds ($\gamma \leq 0.3$). All retrievals converge with a reduced $\chi^2$ lower than 2, confirming the good quality of the retrieval. In Fig. 3 the scatter plots of OD versus IWP and Liquid Water Path (LWP) (left panels), the effective diameters of ice crystals and water droplets (central panels), the ice fraction and the cloud temperature (right panel)

are reported for ice (black), water (red) and mixed (green) clouds. The IWP is derived from the retrieved parameters by using the equation (Yang et al., 2005):

$$\text{OD}_{\text{FIR},\nu}^{ice} = \frac{3 \cdot \text{IWP}}{D_{\text{ei}} \rho_i} \frac{\langle Q_{\text{ei}} \rangle_{\text{FIR},\nu}}{2} \tag{11}$$



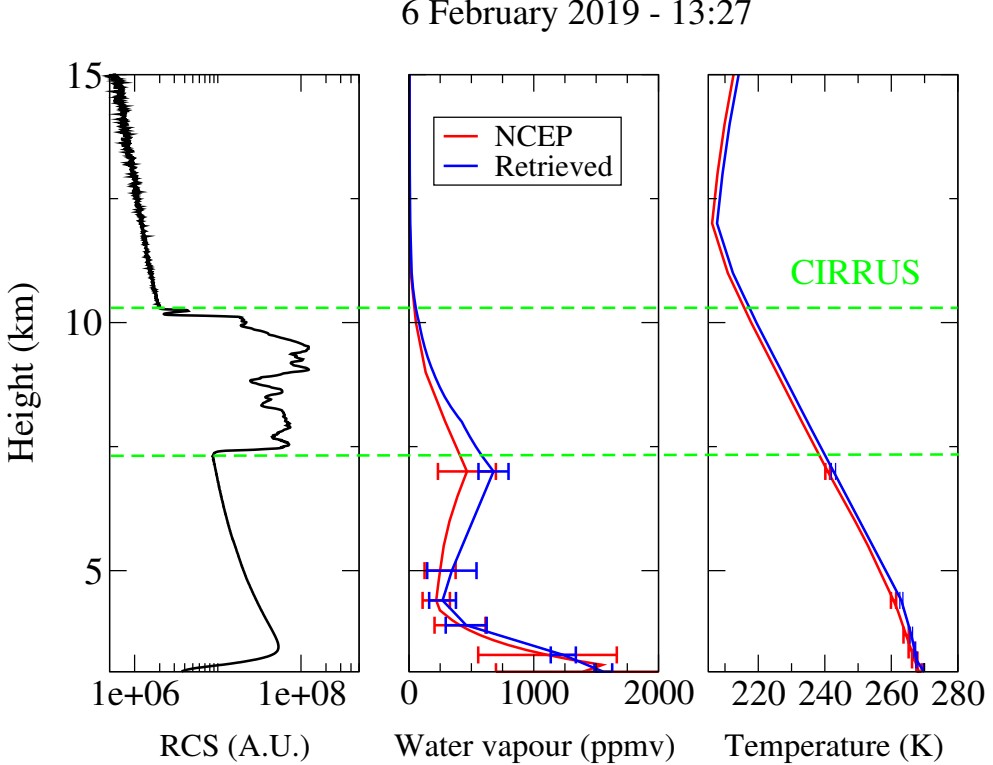

**Figure 2.** In the left panel the LiDAR Range Corrected Signal (RCS) as a function of the height above the ground corresponding to the measurement of Fig. 1. In the middle and right panels the NCEP profiles (red curves) of water vapour volume mixing ratio (vmr) and temperature together with the ones retrieved from FIRMOS data (blue curves).

which comes from eq. (4) for the ice only case, when $Q_e$ can be considered constant and equal to 2 at visible wavelengths because of the large size parameter ($\frac{\pi D_e}{\lambda}$), $\rho_i = 917$ kg m$^{-3}$ is the density of the pure ice and OD$_{\mathrm{FIR}}^{ice}$ is referred to the ice
component. From the comparison with eq. (4), the OD$^{ice}$ at visible wavelengths depends on the IWP and $D_{ei}$. The LWP can be derived since eq. (11) is also valid for water case by replacing the parameters for ice with the analogue ones for water and using $\rho_w = 1000$ kg m$^{-3}$.

The retrieved ODs are found ranging between 0.01 and 1 for ice clouds, mostly between 0.1 and 10 for mixed clouds and between 0.3 and 10 for water clouds. The effective diameters mostly range between 10 and 40 $\mu$m for ice particles and between
2 and 10 $\mu$m for water droplets. The average values of OD$^{ice}$, OD$^{wat}$, D$_{ei}$ and D$_{ew}$ are found equal to 0.13, 0.77, 19 $\mu$m and 6.7 $\mu$m , respectively. The cloud temperature is found lower than 240 K for ice clouds, and mostly in the range below 220 K, corresponding to cirrus clouds above 8 km of height. The mixed and water clouds are warmer with temperature above 240 K and they occur at lower heights, below 6 km.

Following previous works (Heymsfield et al., 2003), a power law in the form OD $= a \cdot$ IWP$^b$ is used to fit the retrieved data
of IWP and OD for ice clouds. From FIRMOS measurements the fit result provides the coefficients $a = (0.173 \pm 0.005)$

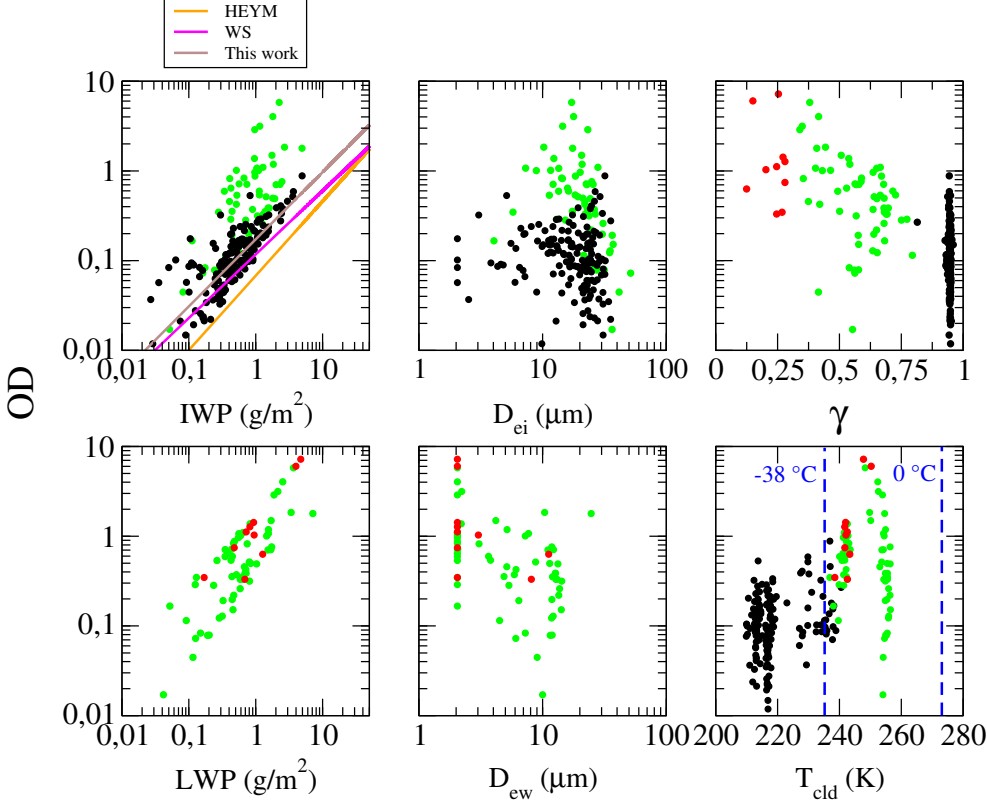

**Figure 3.** On the left panels the scatter-plot of OD vs. Ice Water Path (IWP) and Liquid Water Path (LWP), respectively, for ice (black), water (red) and mixed (green) clouds. The solid blue, orange and magenta lines represent the power-law fit obtained in this work and in the previous works Heymsfield et al. (2003); Wang and Sassen (2002) denoted as HEYM and WS, respectively. In the middle panels the OD vs. effective diameters of ice crystals ($D_{ei}$) and water droplets ($D_{ew}$). In the right upper and lower panels the same scatter-plot of OD vs. the ice fraction ($\gamma$) and cloud temperature ($T_{cld}$).

and $b = (0.748 \pm 0.028)$. The fitted curve is reported in Fig. 3 (in brown). This is close to those found in the other works (Heymsfield et al., 2003) (in orange) and, particularly, in very good agreement with those found by Wang and Sassen (2002) (in magenta) for mid-latitude cirrus clouds.

## 2.2 LiDAR OD measurements and comparison with the retrieved values from FIRMOS

The optical depths of thin cirrus clouds are retrieved from backscatter profiles of the Zugspitze aerosol LiDAR by using the transmittance method introduced by previous works (Young, 1995; Chen et al., 2002; Giannakaki et al., 2007). This instrument (Höveler et al., 2016) transmits laser pulses at 10 W and 100 Hz and it is commonly used for measurements of the stratospheric aerosol load within NDACC (Network for the Detection of Atmospheric Composition Change). The cirrus OD is directly retrieved from the weakening of the LiDAR return $P$ by the cirrus itself. This is principally done, firstly, linearizing of the





LiDAR logarithmic range corrected signal (LRCS) $S = \ln(Pz^2)$, detecting bottom and top of the cirrus layer, defining two
altitude intervals close to the cirrus with pure Rayleigh backscatter, one below the cirrus and one above (typically 1 km) and
calculating a best fit line by least squares for the LiDAR signal in the two intervals. Finally, the OD is given by the half of
the difference between the two fit lines at the upper edge of the cirrus since the light detected by the LiDAR passes the cirrus
twice. This method is suitable for cirrus layers which are optically thin (OD < 1) and which exhibit a significant extinction with
respect to a reasonable signal to noise ratio of the LiDAR signal. At typical altitudes for cirrus in the winter season (8-10 km)
this means an OD roughly larger than 0.01. Another further precondition is a sufficiently low aerosol or cloud interference and
a fairly stable temperature gradient in the two intervals where the fit lines are calculated. Under appropriate conditions directly
determining the transmittance from the LiDAR signal weakening avoids the use of uncertain a priori knowledge. Particularly,
as already mentioned, ice crystals of cirrus clouds exhibit a large variation of shapes and, thus, a large range of ratios between
backscatter and extinction.

As an example, in the upper panel of Fig. 4 is shown the comparison of the ODs retrieved from the LiDAR data through the

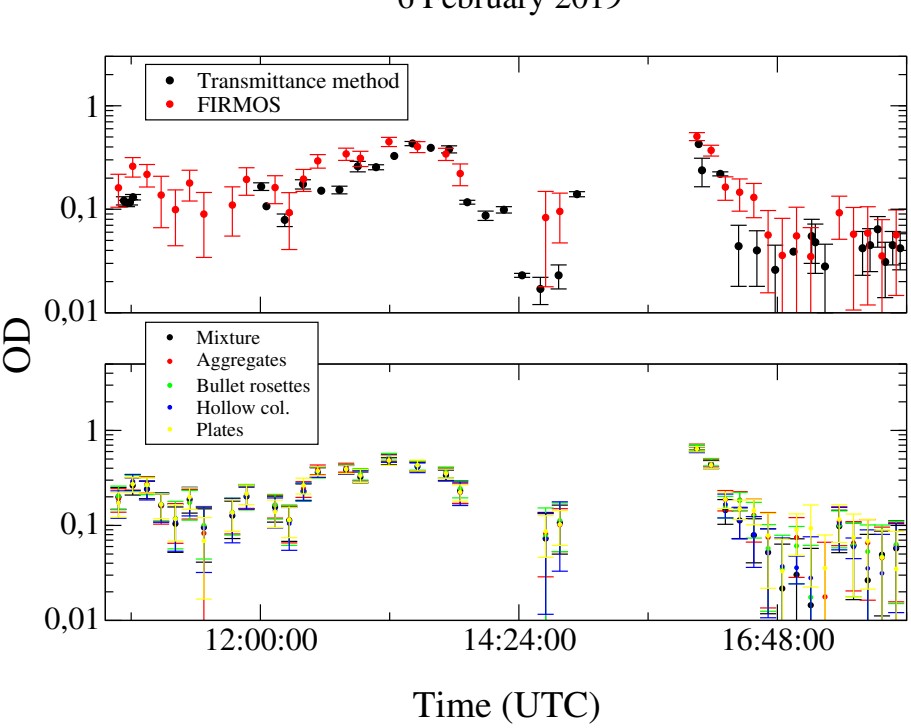

**Figure 4.** In the upper panel comparison of the ODs retrieved from FIRMOS data (in red) on the day 6 February 2019 and those obtained through the transmittance method (in black). In the lower panel the retrieved ODs from FIRMOS by assuming the mixture (black) and different ice crystals habits (red, green, blue and yellow).

transmittance method (black) and those retrieved from FIRMOS data with SACR (red) obtained on the day 6 February 2019 by





using the mentioned mixture of ice crystals habits typical for mid-latitude. The results are in good agreement except for a few cases. These differences are motivated by the fact that the two instruments were not exactly co-located but instead the LiDAR

was measuring at a horizontal distance of 600 m from the FIRMOS spectrometer and 300 m lower. Moreover, the LiDAR has not a field of view but only a beam divergence due to the laser, whereas FIRMOS has a finite one equal to 22.4 mrad, that means the detected signal comes from a wide portion of the observed source. This effect means that the horizontal inhomogeneity of the cloud does not contribute to the LiDAR signal unlike the spectrometer. In order to check whether the habit choice does not affect the OD retrieval, in the lower panel of Fig. 4 the comparison of the ODs retrieved from FIRMOS by assuming different

pristine habits, namely aggregates, bullet rosettes, hollow columns and plates, are also plotted in addition to those obtained by assuming the mid-latitude mixture (in black). The lower panel of Fig. 4 shows that the differences between the retrieved ODs are within the error bars so it can be stated that the retrieval from FIRMOS measurements provides the cloud optical depth regardless of the habit choice used for the retrieval.

The transmittance method approach to retrieve the ODs from LiDAR measurements takes long time if the analysis must be

performed over a large dataset, since it cannot be automatized, and it is not suitable to retrieve OD $> 1$. For this reason the applicability of the Klett method, which can be automatized and, thus, more suitable to analyse large datasets, is also investigated.

## 2.3   OD retrieval by using the Klett method

The retrieval of the OD can be performed by using the Klett inversion method (Klett, 1981, 1985) from the LiDAR backscat-

tering signal $P$ as a function of the altitude $z$. Since only one LiDAR equation and two unknowns, the backscatter ($\beta$) and the extinction ($\sigma$), are involved in this problem, the relationship that binds them has to be assumed. A common solution is to assume a power-law relationship with exponent $k$, called the backscatter-extinction coefficient ratio, as follows:

$$\beta(z) = C \cdot \sigma(z)^k \tag{12}$$

with $C$ constant and $k$ depending on the laser frequency, the aerosol composition and the inhomogeneity of the layers,

particularly the variability of the aerosol size distribution, the particles shape and the mixing with the air, and it also contains the multiple-scattering effect implicitly (Del Guasta et al., 1993; Elouragini, 1995). The coefficient $k$ generally ranges in the interval 0.67–1.00 (Klett, 1981; Elouragini, 1995), but it can also assumes higher and lower values (Takamura and Sasano, 1987; Klett, 1985). Solving the LiDAR equation by using equation 12, the cloud extinction profile ($\sigma_c(z)$) is calculated as a function of the height $z$ as follows:

$$\sigma_c(z) = \frac{\exp[\frac{S(z)-S_r}{k}]}{\sigma_r^{-1} + \frac{2}{k} \int\limits_z^{z_r} \exp[\frac{S(z')-S_r}{k}]dz'} - \sigma_m(z) \tag{13}$$

where $S(z) = \ln(P(z)z^2)$ represents the LRCS, $\sigma_m(z)$ is the molecular extinction derived from the atmospheric profiles, $S_r$ and $\sigma_r$ denote the LRCS and the extinction at a reference height $z_r$ fixed at 500 m above the CTH, respectively, in which





the backscattering contribution is totally due to the molecules. From the extinction the optical depth is obtained as the integral over the clouds thickness:

$$OD = \int_{CBH}^{CTH} \sigma_c(z)dz \tag{14}$$

## 3  Results and discussion

The assumption of the eq. (12) implies that the exponent $k$ must be known, otherwise it must be fitted. As already mentioned, this parameter depends on the optical properties of aerosol in the air (Chan, 2010), which in turn are related to the thermodynamic phase and the shape of the particles. An accurate knowledge of the particle shape can only be assumed for the liquid

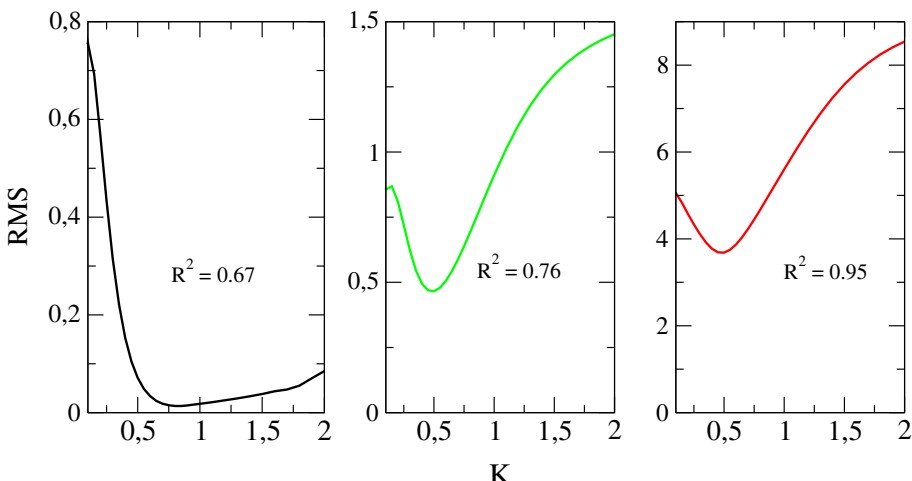

**Figure 5.** Plots of the curves of the root mean square (RMS) differences between the optical depths retrieved from FIRMOS and the LiDAR as a function of the backscatter extinction coefficient ratio k for the ice (black), mixed (green) and water (red) clouds. $R^2$ denotes the correlation index.

phase, in which case they are described in term of spheres, otherwise a complex mixture of different crystal habits must be considered. Anyway, as already shown, the retrieved OD from these measurements does not strongly depend on the habit distribution. For this reason, a possible estimate of $k$ can be obtained by comparing the ODs derived from the LiDAR data (named $OD_{LiDAR}$) by using eq. (13), and those retrieved from FIRMOS spectra (named $OD_{FIRMOS}$). This can be done since the applicability of the Klett method allows to automatize the calculation of the $OD_{LiDAR}$ and to exploit the whole dataset. The best value of $k$ is chosen by taking the minimum of the root mean square (RMS) of the differences between FIRMOS and LiDAR ODs as in previous works (Chan, 2010). Some cases in which the LiDAR signal variations do not correspond to a proportional variation in the FIRMOS spectra were excluded from the comparison, since this indicates that the scenario observed by the two instruments is totally different because of the non perfect co-location. Figure 5 shows the variation of the RMS as a function





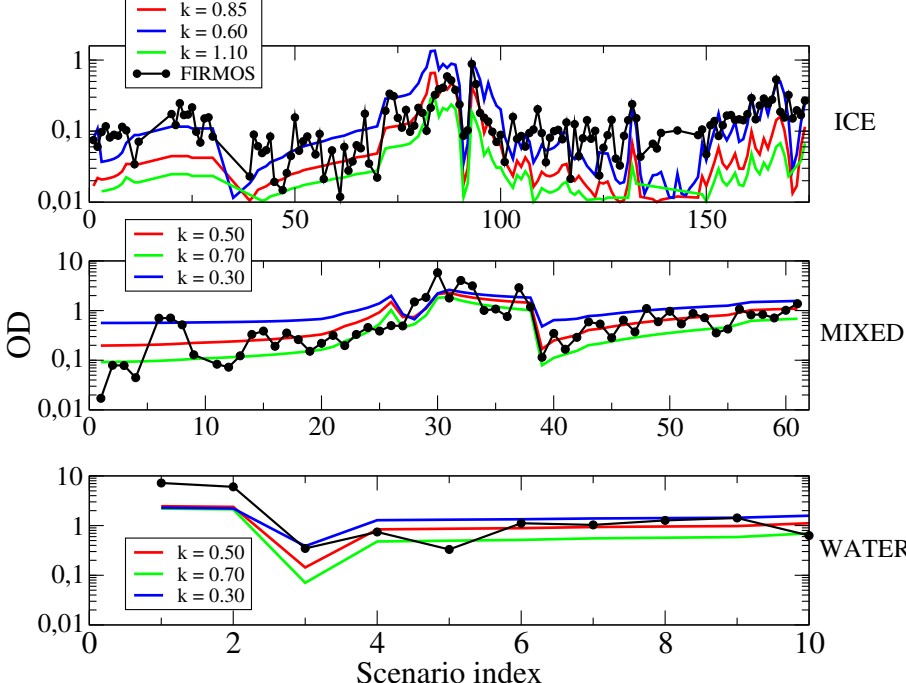

**Figure 6.** In the upper, middle and lower panel the comparison of ice, mixed and water cloud ODs retrieved from FIRMOS (black dots) with those obtained with the Klett approach from LiDAR measurements. The red lines indicate the retrieved OD by using the $k$ of the best fit, while the green and the blue lines show the variability.

of $k$. As $k$ depends on the cloud homogeneity, it is not constant for all scenarios. Thus, it is reasonable to provide a value at

minimum of RMS and a range of variability. The minimum of RMS corresponds to $k$ equal to 0.85 for ice clouds, which is in accordance with the theoretical range of 0.67–1.00, and to 0.50 for mixed and water clouds. The correlation index at the minimum turns out to be 0.67, 0.76 and 0.95 for ice, water ad mixed clouds, respectively, as reported in Fig. 5. The range of variability of the $k$ parameter is estimated as shown in Fig. 6 by using the extreme values corresponding to the blue and green curves that include the most of the scenarios. In case of ice clouds the range of variability is $0.60 \leq k \leq 1.10$, as deducible from

the upper panel of Fig. 6. For mixed and water clouds the range of variability is $0.30 \leq k \leq 0.70$, as shown in the middle and lower panels of the same figure. The differences in the ice clouds scenarios below number 10 and between 139 and 150 (upper panel) are probably due to an inhomogeneity in the cloudy scene observed by FIRMOS and LiDAR, such as the presence of occasional and scattered cirrus clouds during these measurements. The high ODs value (up to 6) of water clouds in the first two scenarios (lower panel) are not reproducible with the Klett approach.

The scatter plot of the $OD_{LiDAR}$ versus $OD_{FIRMOS}$ is reported in Fig. 7 for different values of $k$ and for all the types of clouds. The scatter plot obtained for the $k$ values minimizing the RMS is shown in the central panel. The upper and lower panels show the scatter plots for values of $k$ lower and greater than the minimum, respectively, corresponding to the estimated variability.

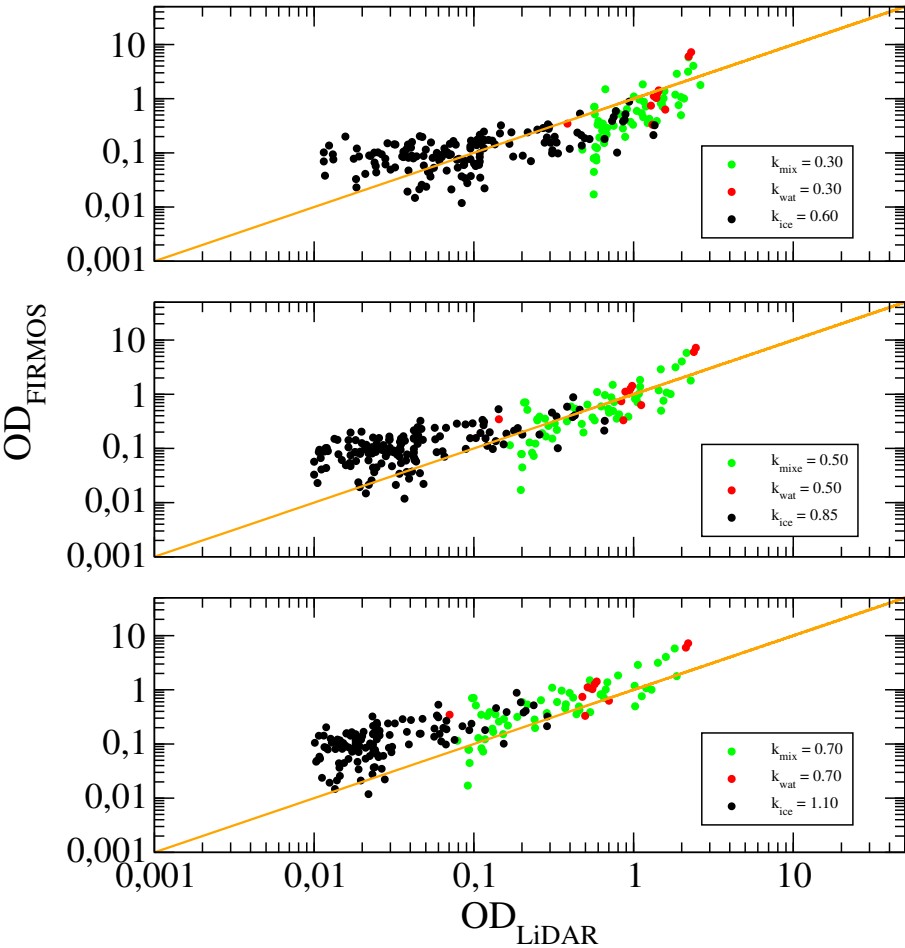

**Figure 7.** In the middle panel the scatter-plot between the optical depths retrieved from FIRMOS and the backscattering LiDAR for $k$ equal to the value at the minimum as plotted in Fig. 5. In the upper and lower panels the same comparison for values of $k$ larger and lower than the minimum, respectively. The black, green and red colors indicate the ice, mixed and water clouds as reported in Fig. 3 and 5.

Once the $k$ value is estimated, this approach allows to rapidly calculate the ODs and the corresponding variability by applying the Klett method to the LiDAR measurements, using a simple algorithm.

## 250 4 Conclusions

In this work the spectral DLR measurements performed by means of the Far-Infrared Radiation Mobile Observation System (FIRMOS) spectrometer installed in the Winter 2018–2019 (between December 2018 and February 2019) at the Alpine observatory on the summit of Mount Zugspitze (Germany), were used to determine the so called backscatter-extinction coefficient ratio for ice, water and mixed-phase clouds. The SACR routine, based on an optimal estimation approach, was used with



FIRMOS spectral radiances, to calculate the cloud optical and micro-physical properties, such as the effective diameters of ice crystals and water droplets and the optical depth, together with the ice fraction and the atmospheric profiles. The different behaviour of the ice and water refractive index below 1000 cm$^{-1}$ was exploited to retrieve the ice fraction by modelling cloud as an homogenous mixture of ice crystals and water droplets, since measurements of depolarization were not available. The single scattering properties of ice crystals were taken from specific databases provided by Ping Yang et al. and the average

values are obtained by assuming a mixture of habits typical for mid-latitude. A statistical parameterization between the IWP and the OD in the form OD $= a \cdot$ IWP$^b$ was used to fit data and coefficients $a$ and $b$ were found in accordance with those derived from previous works.

The simultaneous availability of a backscatter LiDAR, co-located near the Zugspitze summit, allows to constrain the cloud geometrical properties for simulating the radiative transfer in the presence of clouds. The backscattering signal was used to

modulate the cloud vertical distribution of the optical depth and take into account the multiple scattering effect of the different layers. These data also allowed to validate the retrieved optical depth from FIRMOS spectra with those obtained from LiDAR data by applying a transmittance method. Since a power-law among the backscatter and the extinction coefficients is commonly assumed to invert the LiDAR equation, the procedure to estimate the best values of the exponent of this relationship, the backscattering extinction coefficient ratio $k$, is presented and discussed. This approach mainly consists in the minimization

of the root mean square (RMS) differences between the ODs retrieved from FIRMOS spectra and those obtained from LiDAR measurements by using the Klett method. In such a way, it was found that for cirrus clouds $k$ varies in the interval 0.60–1.10, providing a variability of 0.25 with respect to the best value at the minimum of the RMS differences equal to 0.85, while for water clouds $k$ varies in the range 0.30–0.70, with a variability of 0.20 with respect to the best value equal to 0.50. The assessment of $k$ represents an useful information for the application of the Klett method to LiDAR measurements. This approach

opens the possibility to retrieve the cloud ODs from large datasets of LiDAR measurements, which can be used to determine the climatology of this parameter.

*Data availability.* FIRMOS and LiDAR data are available via the ESA campaign dataset website https://earth.esa.int/eogateway/campaigns/firmos (Palchetti et al., 2020a), https://doi.org/10.5270/ESA-38034ee)

*Author contributions.* GDN, conceptualization, designed methodology and prepared the manuscript. FD,MB,SV,LP,GB,HS,RS,AM, run the

instruments during the campaign. GDN,LP,HS,SD,MG,CB, prepared and evaluated L1 data analysis. GDN, HS provided L2 data analysis. LP, responsible for the FIRMOS project. All authors revised the manuscript.

*Competing interests.* The authors declare that they have no conflict of interest.





*Acknowledgements.* The authors gratefully acknowledge the funding support by the European Space Agency with the FIRMOS project
(ESA–ESTEC Contract No. 4000123691/18/NL/LF) and the Italian Space Agency with the research projects SCIEF (Italian acronym of

Development of the National Competences for the FORUM experiment - ASI contract No. 2016-010-U.0). They also thank the Italian PNRA
(Programma Nazionale di Ricerche in Antartide) and specifically the project FIRCLOUDS (Far Infrared Radiative Closure Experiment For
Antarctic Clouds) which allow to fund the contract of the first author.The consolidated dataset is freely available via the ESA campaign
dataset website: https://earth.esa.int/web/guest/campaigns under entry https://doi.org/10.5270/ESA-38034ee.





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
