# Peer review of "Comparison of mid-latitude single and mixed-phase cloud optical depth from co-located infrared spectrometer and backscatter LiDAR measurements"

_Atmospheric Measurement Techniques, 2021_

## Author Comment (AC1)

**Reply to reviewer comments amt-2021-104-RC1**

**General Comments:**

The manuscript is well written and clearly understood. The scientific approach is valid, and the authors present some interesting results. For example, the power law relationship between ice water path and optical depth are in reasonable agreement with previous studies. Furthermore, the values for the exponent, *k*, which is often assumed as unity is relevant for the inversion of lidar data. Measurements of clouds in the far-infrared are needed in advance of satellite missions like the Far infrared Outgoing Radiation for Monitoring and Understanding (FORUM). However, there are some minor suggestions/corrections that should be considered before this work is published. For instance, the use of backscatter-extinction coefficient ratio to identify the exponent, *k*, could be an unnecessary cause of confusion. The small sample of liquid water clouds (only 10 cases) is not very convincing. At the very least, it should be noted that there is the potential that these may be misclassified from mixed phase cases. However, the strong results from the ice phase/cirrus cloud cases supersedes any weaknesses elsewhere, and this manuscript should be published after minor revisions.

We are grateful to the reviewer for the positive and constructive comments and suggestions, which are all considered in the revised paper to improve quality and clarity.

**Detailed Comments:**

Page 1, Line 8: Though the term is used by Chan 2010, backscatter-extinction coefficient ratio does not appear to be widely accepted to describe the power law exponent, *k*, in most literature. In fact, it can be confused with the more commonly used backscatter-to-extinction ratio (or it's inverse, the extinction-to-backscatter ratio) which is used to solve the lidar equation. My suggestion is to avoid using this potentially confusing term throughout the manuscript.

Agree with the reviewer and replaced the ambiguous term with the expression "power-law exponent k" throughout the text.

Page 6, Line 147: The choice of ice water fraction, γ < 0.3, to identify water clouds may result in misiclassification of mixed phase. For example, Turner 2003 uses a cutoff of 0.2 which includes a majority of the 10 liquid water clouds in the sample judging by Figure 3. However, as noted by the authors, without the availability of depolarization measurements it is difficult to determine the vertical distribution of thermodynamic phase. It should be noted, though, that the majority of the liquid water phase is indistinguishable from mixed phase in Figure 3 (OD, LWP, effective diameter, and temperature), with the exception of ice water fraction.

This is a very good suggestion, it is totally true that the choice of the threshold on the ice fraction (γ) could result in a misclassification. We followed this suggestion to reclassify our cases using a cut off of 0.2 as done by Turner 2003 together with the condition that the absolute difference between the retrieved γ and the cut off must be larger than the retrieval error ($\Delta\gamma$) to identify the case as pure water, this in order to avoid any possible misclassification, namely:

water case if : $\gamma < 0.2$ & $|\gamma - 0.2| > \Delta\gamma \rightarrow$ otherwise mixed or ice

so that in this case we are sure that γ is below the threshold regardless of the retrieval error. In this way all previous cases classified as water turn out to be classifiable as mixed.

Figure 3-5-6-7 were modified accordingly. Particularly, in Figure 5 the RMS for mixed clouds was recalculated considering the corrected dataset and the water one was removed. Finally, the text was corrected accordingly to the adjustments.

Page 10, Line 189: Could the differences in wavelength (532 nm vs. FIR) also contribute to the differences in optical depth? If so, how does that affect the other results of the study (e.g. *k* retrieval).

Actually, the optical depth shows a strong spectral dependence only in the infrared spectrum and not at the visible frequencies. The OD in the FIR is related to the OD at visible frequencies through Eq. (4) with the extinction efficiency, which can be assumed equal to 2 in the visibile range because of the large size parameter (Eq. (11)). Following this approach, the OD in the visible (specifically at 532 nm) retrieved from the lidar signal can be directly compared with the OD determined from the spectral measurements in the FIR. Therefore, the causes that produce the differences in the comparison of Figure 4 are mainly due to the not perfect co-location of the instrument (in particular the lidar is installed at a lower height of 300 m and 600 m horizontally) and, the different field of view (FOV) of the two instruments are different, in particular FIRMOS has a 22.4 mrad FOV, which corresponds at 5 km above the instrument (where usually cirrus occur) to a region of about 100 m of diameter, whereas for the lidar is much smaller.

Page 10, Line 193: The sentence beginning with, "In order to check whether the habit choice does not affect the OD retrieval…" could be more clearly expressed as, "In order to check whether the habit choice affects the OD retrieval…"

Corrected in the text.

Page 11, Line 231: How many cases are removed in order to determine the optimal value of *k*? And are such removals necessary for ice, liquid, and mixed phase clouds?

16 measured spectra from the total of 261 were removed regardless of the type of clouds because of inconsistency between the FIRMOS measurements with lidar ones. In fact, in these cases, as you can see from the examples in the figures below (backscattering in the left panels), while the lidar signal decreases the FIRMOS signal increases. These cases denote that the two instruments are clearly observing different scenarios, so they were excluded from the analysis.

[Figure]

[Figure]

[Figure]

 The high liquid water optical depths likely fully attenuate the lidar signal which is why it is not reproducible using the Klett inversion.

Correct. This sentence was also added to the revised text.

Figures 6 and 7: Because *k* is often assumed to have a value of unity, it would be instructive to see how that value compares to the optimal value determined from this study.

We thank the reviewer for this advise, we also added the curves of k = 1.0 in both panels of Figure 6.

---

## Author Comment (AC2)

**Reply to reviewer comments amt-2021-104-RC2**

The work is very interesting and innovative. It would also have been interesting to have high spectral resolution lidar measurements of water vapor and temperature. In general this approach opens the possibility to retrieve the cloud ODs from large datasets of LiDAR measurements, which can be used to determine the climatology of this parameter. This aspect makes the work very important also for future studies.

We are grateful to the reviewer for the positive comments, which well captures the aim of this work. During the campaign, a Raman Lidar system present at the site  was also operated in parallel with the other measurements (dataset can be downloaded from https://fts.fi.ino.it/forum/firmos/zugspitze-dataset/) but only manually for very few (four) thin cirrus cases, which are not relevant for this work.